# Insight Over Sight? Exploring the Vision-Knowledge Conflicts in Multimodal LLMs

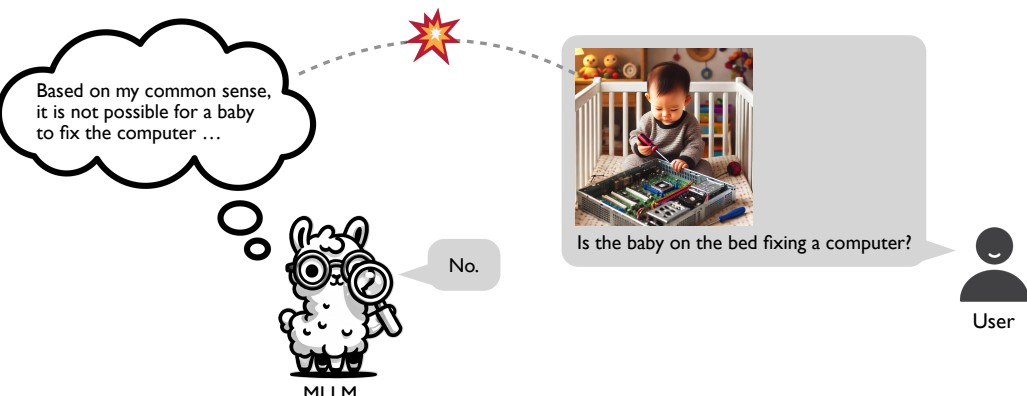

Figure 1: Illustration of the vision-knowledge conflict, where the visual input contradicts MLLM's inherent knowledge. The MLLM's response over-relies on its inherent commonsense knowledge.

## Abstract

This paper explores the problem of commonsense-level vision-knowledge conflict in Multimodal Large Language Models (MLLMs), where visual information contradicts model's internal commonsense knowledge (see Figure 1). To study this issue, we introduce an automated pipeline, augmented with human-in-the-loop quality control, to establish a benchmark aimed at simulating and assessing the conflicts in MLLMs. Utilizing this pipeline, we have crafted a diagnostic benchmark comprising 374 original images and 1,122 high-quality question-answer (QA) pairs. This benchmark covers two types of conflict targets and three question difficulty levels, providing a thorough assessment tool. Through this benchmark, we evaluate the conflict-resolution capabilities of nine representative MLLMs across various model families and find a noticeable over-reliance on textual queries. Drawing on these findings, we propose a novel prompting strategy, "Focus-on-Vision" (FoV), which markedly enhances MLLMs' ability to favor visual data over conflicting textual knowledge. Our detailed analysis and the newly proposed strategy significantly advance the understanding and mitigating of vision-knowledge conflicts in MLLMs. The data and code will be released.

## 1 Introduction

Large Language Models (LLMs) (Brown et al., 2020; OpenAI, 2022; Touvron et al., 2023; OpenAI, 2023a; Meta AI, 2024) have reshaped the landscape of deep learning for their comprehensive capabilities in language understanding, reasoning, and generation (Wei et al., 2022a;b; Chen et al., 2023). This evolution has paved the way for the emergence of Multimodal Large Language Models (MLLMs) (Li et al., 2022a; 2023b; Lyu et al., 2023; Dai et al., 2023; Liu et al., 2024d; Zhu et al., 2023; Bai et al., 2023; Liu et al., 2024b; Wang et al., 2024; OpenAI, 2023b; Liu et al., 2024c; Tong et al., 2024; Gemini Team, 2024; OpenAI, 2024b), which integrate a vision model with the LLM to process and respond to visual information. MLLMs like `GPT-4o` (OpenAI, 2024b) and `LLaVA-NeXT` (Liu et al., 2024c) have exhibited remarkable proficiency across various vision-language tasks such as

image captioning (Chen et al., 2015), visual question answering (VQA) (Antol et al., 2015), and visual reasoning (Johnson et al., 2017; Yue et al., 2024).

However, the persistence of knowledge conflicts in LLMs remains a significant challenge for MLLMs, often cited as a primary cause of hallucinations in these systems (Liu et al., 2023a; Guan et al., 2024). Introducing visual information into MLLMs generates a novel form of discrepancy termed "**vision-knowledge conflict**", where the visual data contradicts the model's pre-existing parametric knowledge. Prior research has mainly focused on assessing and comprehending these conflicts through the lens of **factual knowledge** by generating counterfactual images via image editing techniques (Guan et al., 2024). Nonetheless, the aspect of vision-knowledge conflict related to **commonsense knowledge**,[1] which proves more challenging for LLMs and MLLMs due to its implicit and nuanced nature (Li et al., 2022b), remains under-investigated.

To address this gap, we propose an automated pipeline with human-in-the-loop to develop a benchmark for simulating and analyzing commonsense-level vision-knowledge conflicts in MLLMs. Illustrated in Figure 2, our pipeline consists of four key modules: knowledge component extraction, counter-commonsense query construction, image generation, and question-and-answer (QA) generation. This framework streamlines the generation of counter-commonsense inputs from scratch and is modularly designed to facilitate the addition of further conflict categories and QA formats in the future. We demonstrate the application of our framework by developing the CONFLICTVIS benchmark built on the Open Mind Common Sense dataset (Singh et al., 2002). The benchmark focuses on the critical aspects of commonsense knowledge in the visual scene: Subject, Action, and Place. It consists of **374** original images with **1,122** high-quality QA pairs spanning two conflict targets and three question difficulty levels, all verified by human annotators.

Using the crafted benchmark, we evaluate nine representative MLLMs from five model families, providing insights into model behaviors, the causes of conflicts, and effective methods designs to mitigate the negative effects of conflicts. Notably, when facing knowledge conflicts, MLLMs tend to over-rely on their parametric knowledge for simpler questions. For instance, the top commercial model, `Claude-3.5-Sonnet`, exhibits a memorization ratio (MR) on parametric knowledge of 43.6% on Yes-No questions, substantially higher than results on more complex subjective questions; the largest open-source model evaluated, `LLaVA-NeXT-34B`, shows a MR of 26.7% on Yes-No questions, three times more than that on subjective questions. Our detailed analysis of the failure cases reveals that MLLMs generally focus more on the textual query than on the visual context. Drawing on these observations, we revisit several existing training-free methods (Leng et al., 2024; Liu et al., 2024f; Wei et al., 2022b) to enhance the impact of visual context in answer generation. Interestingly, although Chain-of-Thought prompting (Wei et al., 2022b) improves model's reasoning abilities, it directs MLLMs to over-utilize the parametric knowledge in the reasoning process, leading to inaccurate conclusions or refusals. From this understanding, we propose **Focus-on-Vision** (FoV) prompting to instruct MLLMs to prioritize visual information, which markedly improves the performance.

Our main contributions are summarized as follows:

- We introduce an innovative framework to automatically construct counter-commonsense benchmarks from the ground up. This platform allows the flexible definition of conflict categories and QA formats, facilitating the large-scale creation of conflict samples with minimal human effort.

- We present CONFLICTVIS , a pioneering diagnostic benchmark specifically designed to evaluate vision-knowledge conflicts at the commonsense level in MLLMs. The benchmark is meticulously validated by human experts to guarantee the quality of data.

- We perform extensive experiments on nine representative MLLMs, uncovering critical insights into model performance and the underlying mechanisms of the conflict. Leveraging these insights, we propose a novel Focus-on-Vision (FoV) prompting as an effective solution to significantly minimize vision-knowledge conflicts in MLLMs.

## 2 CONFLICTVIS BENCHMARK

This section outlines the pipeline and the data utilized to construct the CONFLICTVIS benchmark.

---

[1]To illustrate, a factual knowledge example is "The Eiffel Tower is 330 meters tall", while a commonsense knowledge example is "Babies cry when they are hungry".

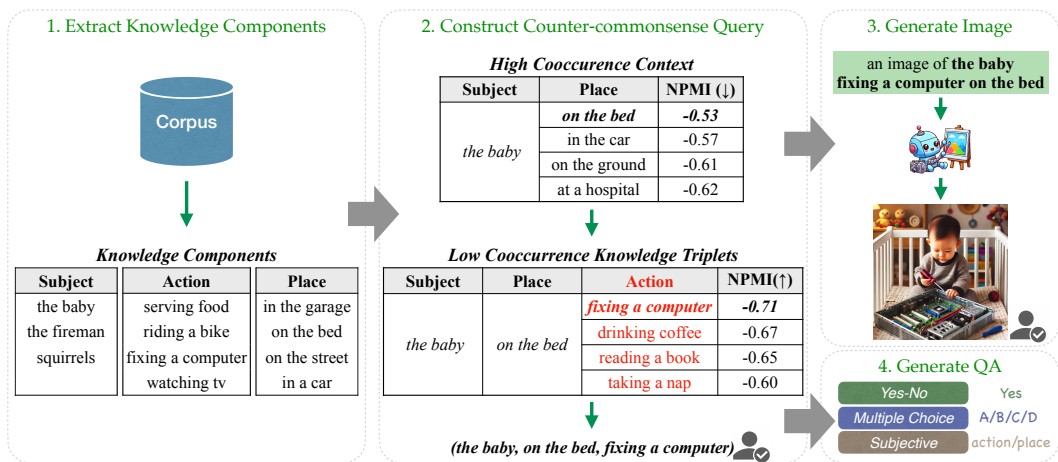

Figure 2: An automated pipeline with human quality control to construct images and question-answer pairs that conflict with commonsense knowledge.

## 2.1 AUTOMATED PIPELINE

Commonsense knowledge refers to the information generally accepted by the majority of people about everyday life, encompassing practical principles on how the world functions (Singh et al., 2002). Inspired by the widely-used image captioning dataset (Chen et al., 2015), we explore a specific type of commonsense knowledge encapsulated as a triplet $< s, a, p >$, where $s$ is the Subject, $a$ is the Action, and $p$ is the Place of the action or subject. For example, the statement "a baby (Subject) drinking water (Action) on the bed (Place)" illustrates this format. In this structure, the Subject outlines the main object's appearance, the Action describes the primary activity and its interactions with other relevant objects, and the Place highlights the background objects and setting. This three-part framework efficiently captures the vital details that are accurately reflected in an image.

We then describe the strategies for creating images and corresponding QA pairs that contradict the commonsense knowledge based on an input corpus. In general, we construct triplets containing low co-occurring concepts to serve as the counter-commonsense queries for image generation and fill pre-defined templates to generate the QA pairs. The pipeline contains the following four steps:

**1. Extract Knowledge Components from Input Corpus.** The process commences with the extraction of knowledge components from the corpus, which contains human commonsense knowledge in natural language. Drawing inspiration from Li et al. (2021) on the creation of a compositional generalization test set, we identify frequently occurring components in the corpus and enumerate all possible combinations to generate knowledge triplets. To this end, we employ a transformer model pipeline to extensively annotate the syntactic labels, including dependency (DEP), Part-of-Speech (POS), and Named Entity (NE). Table 3 in Appendix lists the linguistic rules to identify each component:

- **Subject Phrases**: Our framework examines noun chunks, selecting those where the head noun functions as the nominal subject, indicated by dependency labels "nsubj" or "nsubjpass".
- **Action Phrases**: The process starts by identifying the verb token ("VERB") and then examines its dependents for direct objects marked as "dobj". Upon finding such objects, our framework includes any determiners or compound modifiers present to form the complete action phrase. In addition, verbs are converted to the "VBG" form to align with the common practice in image captions.
- **Place Phrases**: Distinguishing place phrases from other prepositional phrases relies on more than just syntactic labels. Accordingly, the framework employs semantic role labeling techniques by utilizing a BERT-based model (Gardner et al., 2017) to annotate each word's semantic role. It then extracts the phrase with a location tag ("ARGM-LOC") that contains 3 to 4 words.

To develop a high-quality benchmark, our framework filters the top subject ($N_S$), action ($N_A$), and place ($N_P$) phrases, respectively, based on the occurrence. It also removes named entities from the

subject and place phrases and merges similar phrases (e.g., "a doctor" and "the doctor") by keeping the most frequently occurring variant.

**2. Construct Counter-commonsense Query.**   We aim to construct scenes with a single anomalous concept (i.e., target) that seldom co-occurs with the other concepts (i.e., the context pair). For instance, the Action component "fixing a computer" and the context pair ("the baby", "on the bed") results in the triplet "the baby on the bed fixing a computer", which contradicts commonsense knowledge, as it is highly improbable for a baby to fix a computer in reality. Specifically,

- The context pair should exhibit a **high co-occurrence** level, acting as a common background.
- The target should display a **low co-occurrence** with the given context, representing the anomaly.

To find the high co-occurrence context pairs, our framework first groups context pairs by their target category, i.e., **Context**$_{Action}$ consists of (Subject, Place) pairs and **Context**$_{Place}$ comprises (Subject, Action) pairs. These groups help to develop the focus of the questions in the next section. We omit **Context**$_{Subject}$ to avoid the ambiguity and potential ethical issues for judging a subject's identity.

We demonstrate how to construct queries with counter-commonsense **Actions**, and a similar approach can be applied to Places. For each subject, we pair it with all the Place components to establish context candidates and select the contexts based on the co-occurrence level. We use Normalized Pointwise Mutual Information (NPMI) to evaluate the co-occurrence of the context pairs $\mathbf{C} = (C_X, C_Y)$:

$$\text{PMI}(C_X; C_Y) \equiv \log_2 \frac{P(C_X, C_Y)}{P(C_X)P(C_Y)}, \qquad \text{NPMI}(C_X; C_Y) \equiv \frac{\text{PMI}(C_X; C_Y)}{-\log_2 P(C_X, C_Y)}. \qquad (1)$$

A higher NPMI score indicates a more frequent co-occurrence of the two context components. For each subject, we select Top-$K$ places with **highest** NPMI scores to form context pairs.

For each selected context pair, we construct counter-commonsense triplets by selecting a target Action component that is not compatible with the context pair. Concretely, we also use the NPMI metric (i.e. NPMI($T$; $\mathbf{C}$)) to evaluate the co-occurrence of the target $T$ and context pair $\mathbf{C}$. For each context pair, we select Top-$M$ actions with **lowest** NPMI scores to form counter-commonsense queries.

To better match real-world knowledge, we leverage an LLM trained on large-scale web data to calculate the probability $P(\cdot)$. For example, to calculate the joint probabilities for multiple co-occurring components, our framework forms a concatenated expression of the concept phrases (e.g., "a baby on the bed" or "a baby on the bed fixing a computer") and inputs it into a pre-trained LLM to obtain the generative probability. Utilizing an LLM here offers an advantage over frequency counting for addressing the issue of zero frequencies for rare concept compositions, which often arises with limited dataset size. Since large language models are trained on large-scale data to learn the statistical tendencies of human language, their generative probability provides an approximation of the likelihood of certain phrase appearing in reality. For the probability of each individual component, we use its relative frequency within its category (Lin, 1999). Following generation, a manual review is conducted here to ensure the data quality (See Appendix B for further details).

**3. Generate Image.**   We leverage state-of-the-art text-to-image model DALL·E 3 (OpenAI, 2024a) to generate image based on the counter-commonsense query. Specifically, we concatenate the components of each triplet in order and prefixing them with "an image of". As shown in Figure 2, the sample prompt for image generation is "an image of the baby fixing a computer on the bed". After image generation, a human annotator is responsible for the quality control, filtering out the low-quality images that are distorted or misaligned with the input prompt. We present detailed filtering guidelines and illustrative examples in Appendix B.

**4. Generate Questions.**   As illustrated in Figure 3, with the constructed counter-commonsense triplets, the pipeline generates questions across three difficulty levels, from easy to hard:

- **Yes-No Question**: For counter-commonsense Actions and Places, our framework utilizes templates "Is/are [Subject Place] [Action]?" and "Is/are [Subject Action] [Place]?", respectively. The correct response to this type of question is typically "Yes".
- **Multiple-Choice Question**: For these questions, the templates include "What is/are [Subject] doing [Place]?" and "Where is/are [Subject] [Action]?" for counter-commonsense Action and

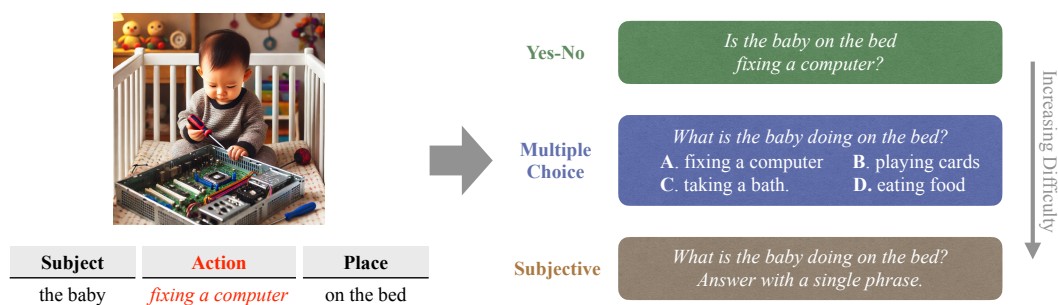

Figure 3: Illustration of generating questions across three difficulty levels, from easy to hard based on the conflict target (*Action* in this example.)

Place, respectively. For a predetermined option count $m$, our framework selects $(m-1)$ seemingly relevant but incorrect targets from the candidates. To this end, the framework partitions candidates into $m-1$ bins $\{B_1, ..., B_{m-1}\}$ based on their NPMI scores relative to the context pair. We randomly sample one candidate from each bin, together with the counter-commonsense target, to formulate the question's multiple-choice options.

- **Subjective Question**: The question text in the multiple-choice question is reused, but no list of options is included. To guide the model toward generating a brief and specific response, we add the constraints like "Answer with a single phrase", with slight adjustment according to the specific prompt for different MLLMs.

## 2.2 CONFLICTVIS BENCHMARK CONSTRUCTION

We introduce how to use the automated pipeline to construct the CONFLICTVIS benchmark. We utilize top 100K sentences from the Open Mind Common Sense (OMCS) dataset (Singh et al., 2002) as our knowledge base. Developed by the MIT Media Lab, OMCS is a rich collection of human commonsense knowledge, consisting of assertive sentences contributed by thousands of

Table 1: Statistics of the constructed benchmark.

| Target | #Triplets | #Images | #QAs |
|--------|-----------|---------|------|
| **Action** | 188 | 171 | 513 |
| **Place** | 156 | 203 | 609 |
| **Total** | 344 | **374** | **1122** |

volunteers. From this dataset, we extract the 100 most frequent Subject phrases ($N_S = 100$), along with 150 most frequent phrases for Actions and Places ($N_A = N_P = 150$). Using feedback from the LLM Vicuna-1.5-13b,[2] we select the top 3 phrases with the highest NPMI scores to create context pairs for each subject ($K = 3$). In addition, we choose the top 3 targets with the lowest NPMI scores to assemble the candidate set of triplets ($M = 3$). After manually filtering out unexpected combinations, the remaining triplets are used for image generation. The low-quality images are again manually removed. The specific guidelines and results in the quality control process are elaborated in the Appendix B. Following this two-stage generation and filtering process, the statistics of the resulting benchmark are detailed in Table 1. In total, CONFLICTVIS consists of 374 original images with 1,122 QA pairs spanning two conflict targets and three difficulty levels, all verified by human experts. Figure 10 in Appendix shows the instances for the subject "a musician".

## 3 EXPERIMENT

### 3.1 SETUP

**Models**  To explore the behavior of MLLMs when encountering commonsense knowledge conflicts, we perform a comprehensive evaluation on 9 MLLMs with 7 representative open-source MLLMs ranging from 8B to 34B, and 2 state-of-the-art commercial MLLMs. This evaluation covers the following five model series: LLaVA (8B, 13B, 34B) (Liu et al., 2024b;c), BLIP-2 (12.1B,

---

[2]https://huggingface.co/lmsys/vicuna-13b-v1.5

13B) (Li et al., 2023b; Dai et al., 2023), Qwen-VL (9.6B) (Bai et al., 2023), GPT-4o (OpenAI, 2024b) and Claude-3.5-Sonnet (Anthropic, 2024).

**Evaluation**   We use Accuracy and Memorization Ratio (MR) (Longpre et al., 2021) as the main evaluation metrics in our experiment. Both metrics require classifying the model's responses into different categories. For Yes-No and Multiple Choice questions, where there is a unique answer, we use exact matching for classification. For Subjective questions, due to the open-ended nature and the complexity of the task (i.e., evaluating both textual and visual relevance and correctness), we rely on human annotators to perform the classification (Guidelines and details are included in Appendix C).

## 3.2   SANITY TEST

**Do the instances in CONFLICTVIS really conflict with MLLMs' commonsense knowledge?**   Before evaluating MLLMs' performance on CONFLICTVIS , we first verify whether the counter-commonsense cases in CONFLICTVIS genuinely create conflicts with the models' commonsense knowledge. To this end, we use the textual inputs in CONFLICTVIS and query the model with the following prompt format:

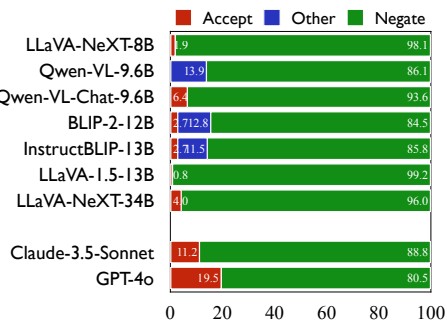

Figure 4: Responses to textual inputs.

```
Based on common sense, is it possible for
[context] [target]?
```

where the context and target are filled with the specific phrases from the input. For example, the query for the instance in Figure 3 is "*Based on common sense, is it possible for the baby on the bed fixing a computer?*" The model's responses are categorized as "negate" (indicating a conflict), "accept" (indicating no conflict), and "other" (where the model does not give a direct answer). The results, shown in Figure 4, demonstrate that the vast majority of cases in CONFLICTVIS indeed present valid knowledge conflicts that can be identified by MLLMs, with conflict rates ranging from 80.5% for GPT-4o to 99.2% for LLaVA-1.5-13B.

**Is CONFLICTVIS a more challenging benchmark for MLLMs?**   Unlike the traditional VQA benchmarks, where visual information typically aligns with the model knowledge, our CONFLICTVIS presents conflicts between these two sources of knowledge, making it a more challenging benchmark for MLLMs. To empirically validate our claim, we conduct a comparative analysis of model uncertainty between our CONFLICTVIS and the conventional VQA benchmark (Antol et al., 2015). A higher degree of uncertainty in model predictions on a particular benchmark indicates that the benchmark poses a greater challenge for the model to navigate. Figure 5 shows the results, where higher entropy value denotes more uncertainty. It is evident that the model's predictions are more uncertain on CONFLICTVIS than the standard VQA benchmark, highlighting the increased challenge posed by our CONFLICTVIS benchmark.

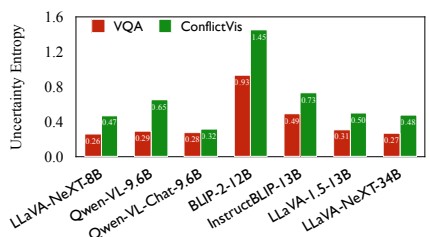

Figure 5: Model uncertainty on traditional VQA dataset and on our benchmark.

## 3.3   BENCHMARKING MLLMS

In this section, we assess how well the MLLMs handle conflicts between vision information and parametric knowledge using our CONFLICTVIS benchmark. Following Longpre et al. (2021), we compare model predictions (i.e., responses) with and without the inclusion of the image and then quantify the instances where model's predictions are influenced by the provided image:

- Align with its predictions without the given image (**Knowledge**, $P_K$): This outcome suggests that the model generates its answer based on its pre-existing parametric knowledge rather than the visual information presented.

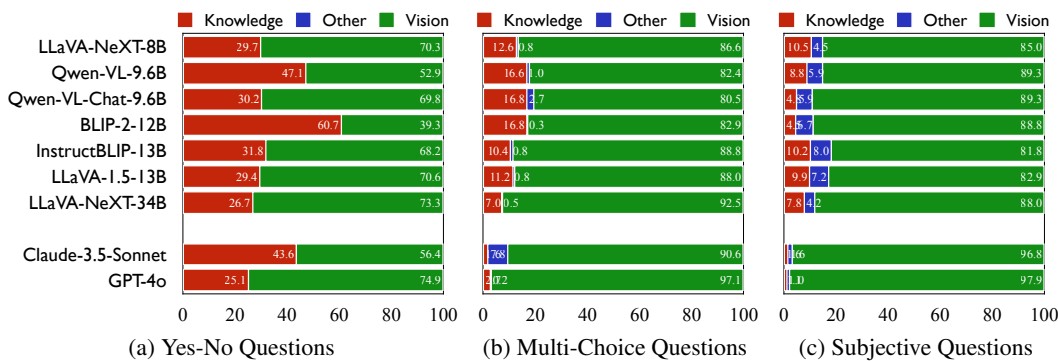

Figure 6: MLLMs' response distributions on different question types. (Responses in Vision category are considered correct.)

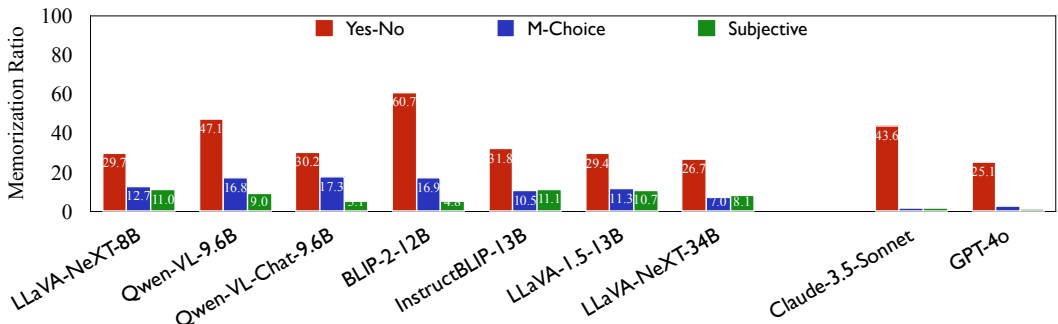

Figure 7: Memorization ratio (MR) of MLLMs, where higher MR values indicate greater model overstability on parametric knowledge.

- Yield a correct answer that aligns with the visual information (**Vision**, $P_V$): This indicates that the model is capable of adapting to new information, adjusting its output to match the visual input.

- Result in a different answer altogether (**Other**): This demonstrates that the model can modify its output based on visual input, though the result does not perfectly align with the visual information.

Figure 6 presents the results. Ideally, the model should output the Vision answer, supported by visual information, rather than the Knowledge answer derived from textual training, or any Other answers. However, both open-source and commercial MLLMs often revert to producing the Knowledge answer, ignoring the visual information, to varying degrees. In general, commercial MLLMs perform better than the open-source counterparts, particularly in handling subjective questions. Among the open-source MLLMs, `LLaVA-NeXT-34B`, being the largest model, delivers the best overall performance, achieving an average prediction accuracy of 84.6%. The top commercial model, `GPT-4o`, reaches an average prediction accuracy of 89.9%.

**MLLMs are more likely to overly rely on parametric knowledge for simpler questions.** One interesting observation from Figure 6 reveals that MLLMs exhibit poorer performance on straightforward Yes-No questions. Notably, the open-source `BLIP-2-12B` achieved a low accuracy of 39.3%, while the commercial `Claude-3.5-Sonnet` managed only 56.4%.

We leverage Memorization Ratio (MR) as a metric to empirically estimate the model's *overstability* — it's brittleness to changing information (Longpre et al., 2021):

$$MR = \frac{P_K}{P_K + P_V}. \tag{2}$$

Figure 7 illustrates the MR across different question types. Notably, the MR values for Yes-No questions are significantly higher than for the other two question types across all MLLMs, indicating that MLLMs tend to overly rely on parametric knowledge when answering simpler questions.

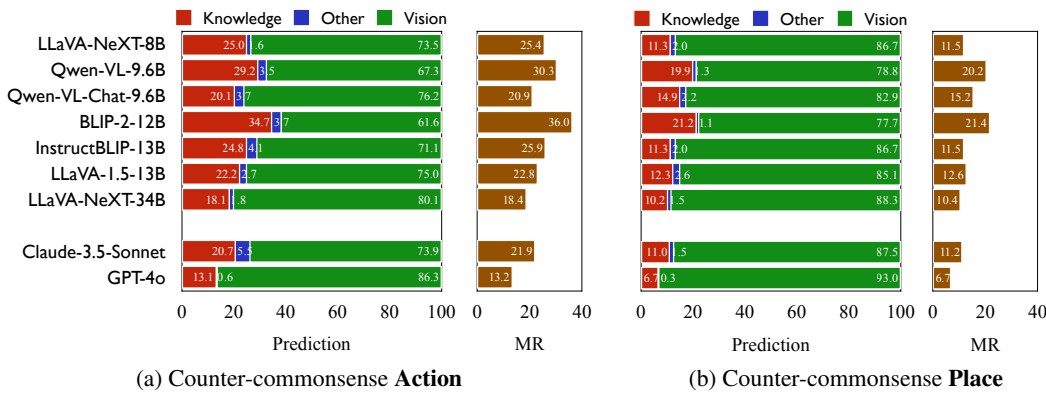

Figure 8: MLLMs' response distributions on two distinct categories of conflict targets.

**Counter-commonsense actions are more challenging for MLLMs.** Our CONFLICTVIS evaluates two distinct types of conflict targets: counter-commonsense actions and places. As shown in Figure 8, the results show that counter-commonsense actions pose more of a challenge than places do. Specifically, the average prediction accuracy for counter-commonsense actions stands at 73.9%, significantly lower than the 85.2% recorded for places. Similarly, the MR for actions is markedly higher at 23.9%, compared to 13.4% for places. This discrepancy could be attributed to the richer visual context available for identifying places (e.g., "in a studio") as opposed to the sparse cues for actions (e.g., "paddling the boat").

## 3.4 STEERING MLLMs TOWARDS VISION

**Underutilization of visual information.** Our primary experiment demonstrates that when MLLMs encounter visual inputs that contradict their pre-existing knowledge, around 18% of the responses generated are not aligned with the visual context but rather reflect the model's prior knowledge. To further understand this discrepancy, we analyze the overall input-output relevancy inside MLLMs (Stan et al., 2024), seeking insights that could guide improvement strategies. Figure 9 illustrates a common case in which the model LLaVA-1.5-13B failed to accurately respond to a subjective question, high-

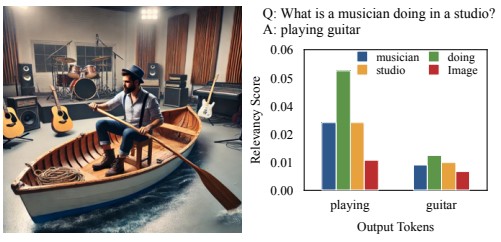

Figure 9: Input-output relevance in a failure case where visual information is underutilized.

lighting the challenge. The model pays more attention to the textual query than to the visual context. Accordingly, the attended tokens {"musician", "doing", "studio"} guide the model to generate the answer "playing guitar" based on the parametric knowledge.

**Effectiveness of Existing Methods.** Based on the above observation, one straightforward improvement strategy is to enhance the impact of visual context when generating answers. We consider several training-free approaches along this direction and propose a novel prompting method:

- *Visual Contrastive Decoding* (Leng et al., 2024) contrasts the output distributions derived from original and distorted visuals to ensure the generated content closely adheres to the visual inputs.

- *Pay Attention to Image* (Liu et al., 2024f) adaptively adjusts and amplifies the attention weights assigned to image tokens, thereby giving greater prominence to visual elements.

- *Chain-of-Thought (CoT)* prompting (Wei et al., 2022b) enables complex reasoning capabilities through intermediate reasoning steps.

While Wei et al. (2022b) demonstrates that CoT can enhance the commonsense reasoning capabilities of LLMs, its applicability to MLLMs in leveraging visual elements under conflict scenarios may not

Table 2: MLLMs' accuracy under different improvement methods.

| Model | Y-N | MC | Subj. | Avg. |
|---|---|---|---|---|
| Qwen-VL-Chat-9.6B | 69.8 | 80.5 | **89.3** | 79.9 |
| + Visual Contrastive Decoding (Leng et al., 2024) | **82.4** | 79.9 | 85.6 | 82.6 |
| + CoT Prompting (Wei et al., 2022b) | 79.7 | 65.8 | 77.8 | 74.4 |
| + FoV Prompting (*this work*) | **82.4** | **83.2** | 87.4 | **84.3** |
| LLaVA-1.5-13B | 70.6 | 88.0 | 82.9 | 80.5 |
| + Visual Contrastive Decoding (Leng et al., 2024) | 72.7 | 89.3 | 84.2 | 82.1 |
| + Pay Attention to Image (Liu et al., 2024f) | **85.6** | 88.8 | **86.1** | **86.8** |
| + CoT Prompting (Wei et al., 2022b) | 38.0 | **89.8** | 76.7 | 68.2 |
| + FoV Prompting (*this work*) | 82.9 | 89.0 | 81.8 | 84.6 |
| LLaVA-NeXT-34B | 73.3 | **92.5** | 88.0 | 84.6 |
| + CoT Prompting (Wei et al., 2022b) | 43.6 | 87.2 | 72.5 | 67.7 |
| + FoV Prompting (*this work*) | **85.8** | **92.5** | **89.8** | **89.4** |
| GPT-4o | 74.9 | 97.1 | 97.9 | 89.9 |
| + CoT Prompting (Wei et al., 2022b) | 66.0 | **98.7** | 93.6 | 86.1 |
| + FoV Prompting (*this work*) | **75.9** | 96.5 | **98.9** | **90.5** |

be directly transferable. In response to this problem, we propose a simple and effective prompting technique named **Focus-on-Vision** (FoV) prompting to explicitly instruct MLLMs to prioritize visual information, which can be seamlessly incorporated into current MLLMs:

```
[textual query] Please focus on the visual information.
```

Table 2 shows the results. We adhere to the methodologies outlined in the original papers by implementing Visual Contrastive Decoding (VCD) and Pay Attention to Image (PAI) techniques on the Qwen-VL-Chat-9.6B and LLaVA-1.5-13B models. Both approaches significantly improve the models' performance, aligning with the outcomes reported in the literature for enhancing MLLMs trustworthiness. Interestingly, we notice that CoT prompting diminishes the performance of the models. Upon examining the failure cases, we find that though CoT envokes the commonsense reasoning abilities of MLLMs, these models can over-rely on their parametric knowledge to draw the conclusion, which results in an increased number of inaccurate response or refusals. We present the corresponding failure cases in Appendix E.2. In contrast, our Focus on Vision (FoV) prompting technique addresses this issue by guiding MLLMs to prioritize visual information, which boosts performances across models.

## 4 RELATED WORK

**Knowledge Conflicts.** Knowledge conflicts in LLMs can be divided into three categories (Xu et al., 2024b): within the retrieved context (Chen et al., 2022), within model's parametric knowledge (Huang et al., 2023), and between the context and the model's parametric knowledge (Xie et al., 2024; Wu et al., 2024a). These conflicts can lead to incorrect or inconsistent responses, undermining the model's trustworthiness (Xu et al., 2024b; Xie et al., 2024). In MLLMs, these conflict types expand to multimodal inputs, where conflicts can occur between the input image and the text instruction (Liu et al., 2024e; Han et al., 2024), or when the image contains counterfactual information that contradicts the model's parametric knowledge (Liu et al., 2024e; Guan et al., 2024).

To evaluate the conflicts between visual information and the parametric knowledge in MLLMs, HallusionBench (Guan et al., 2024) manually edits collected informational graphics to create normal-counterfactual image pairs, and verifies model consistency with simple Yes-No questions. AutoHallu-sion (Wu et al., 2024b) introduces an automated approach to generate counterfactual scenarios by altering correlated objects in images, utilizing Yes-No questions to probe object existence and spatial relationships. In the context of commonsense knowledge, PhD (Liu et al., 2024e) generates counter-

commonsense images through manual collection and synthesis, employing Subjective questions for assessment. Our proposed CONFLICTVIS distinguishes itself by:

- Incorporating a wider range of question types and conflict targets, including Yes-No, Multiple-Choice, Subjective questions and counter-commonsense actions and places. This approach allows for a more thorough evaluation of MLLMs. Our findings indicate that MLLMs tend to disregard visual information more frequently when responding to simpler questions and conflicting action is more challenging for MLLMs to resolve.

- Automatically generating counter-commonsense images based on the co-occurrence of knowledge components and crafting questions around these counter-commonsense aspects, ensuring better image-question alignment. With the aid of advanced text-to-image models, our framework can generate instances tailored to specific needs.

**Hallucination in MLLMs.** Hallucination in MLLMs refers to cases where the model generates descriptions that conflict with the given visual context. MLLMs' hallucinations are typically categorized based on the type of incorrect information, such as non-existent objects, incorrect object attributes, and inaccurate object relations (Liu et al., 2024a). Relevant research has mainly focused on two key areas: developing benchmarks and metrics to detect hallucinations (Rohrbach et al., 2018; Li et al., 2023c; Liu et al., 2023a; Wang et al., 2023) and proposing strategies to mitigate them (Liu et al., 2023a; Leng et al., 2024; Huang et al., 2024; Liu et al., 2024f).

In the context of knowledge conflicts, hallucination occurs when the model gives precedence to its intrinsic knowledge rather than the visual context (Liu et al., 2023a; Guan et al., 2024; Liu et al., 2024e;a). Our study explores this issue by simulating knowledge conflicts that MLLMs may face during inference, and evaluates the effectiveness of different strategies to mitigate hallucinations (Leng et al., 2024; Liu et al., 2024f) by enhancing the model's fidelity to visual inputs. Furthermore, we introduce a novel prompting technique that explicitly instructs MLLMs to prioritize visual information, thereby significantly and consistently reducing the occurrence of hallucinations.

**Benchmarks for MLLMs.** Traditional vision-language benchmarks are designed to assess independent skills, including image captioning (Chen et al., 2015), visual grounding Rohrbach et al. (2016), and visual question answering (Antol et al., 2015; Hudson & Manning, 2019). However, with the emergence of MLLMs, there is a growing need for more comprehensive and tailored benchmarks. The strong zero-shot abilities and advanced language generation skills exhibited by MLLMs can render traditional benchmarks insufficient, as they may not account for the diversity of responses or the full range of MLLM capabilities. To address these limitations, researchers have developed more complex and comprehensive benchmarks to assess MLLMs across a wider range of capabilities (Yue et al., 2024; Fu et al., 2023; Li et al., 2023a; Liu et al., 2023c). Meanwhile, diagnostic benchmarks have also been proposed to evaluate specific challenges or traits in MLLMs, such as hallucination (Li et al., 2023c; Liu et al., 2023a), social bias (Howard et al., 2024), and model safety (Liu et al., 2023b). CONFLICTVIS is a pioneering analytical benchmark to introduce conflicting visual contexts against the model's commonsense knowledge, allowing us to investigate model performance in the face of vision-knowledge conflicts.

## 5 CONCLUSION

This paper introduces an innovative approach to understand and mitigate the significant challenge of vision-knowledge conflicts in MLLMs, with a particular focus on commonsense knowledge. By developing an automated pipeline and the CONFLICTVIS benchmark, we offer a tool for systematically evaluating and understanding the nuances of these conflicts. Our extensive experiments across nine representative MLLMs reveal a tendency toward over-reliance on parametric knowledge, especially in simpler queries, which underscores the importance of the visual context in these models' decision-making processes. Building on these insights, we evaluate three improvement methods and design a new Focus-on-Vision prompting to reinforce the significance of visual input, thereby reducing the impact of conflicts. Our work paves the way for future research aimed at improving the robustness and reliability of MLLMs. Our goal is to ensure these models can effectively integrate and interpret multimodal data, mitigating internal knowledge inconsistencies.

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

# A BENCHMARK DETAILS

## A.1 LINGUISTIC RULES TO EXTRACT KNOWLEDGE COMPONENTS

Table 3: Linguistic rules to identify different components in the knowledge triplet.

| Category | Main Requirement | Example |
|---|---|---|
| Subject | `p in noun_chunks && p.root.dep in ["nsubj", "nsubjpass"]` | a baby, a chef |
| Action | `p[0].pos == "VERB" && p[-1].dep == "dobj"` | fixing a computer |
| Place | `p.srl = "ARGM-LOC" && 3 <= p.length <= 4` | at the bookstore |

## A.2 EXAMPLES IN CONFLICTVIS

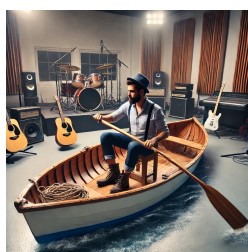 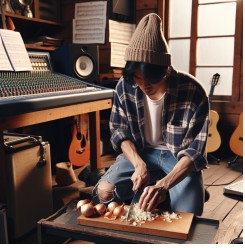 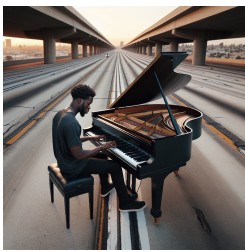 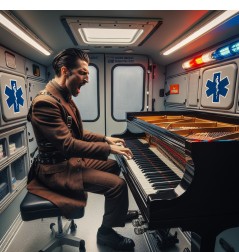

(a) a musician paddling the boat in a studio    (b) a musician chopping some onions in a studio    (c) a musician playing the piano on a freeway    (d) a musician playing the piano in an ambulance

Figure 10: Some examples for the subject "a musician" in the CONFLICTVIS Benchmark.

# B HUMAN QUALITY CONTROL FOR CONSTRUCTING CONFLICTVIS

This section outlines the quality control procedures within our automated pipeline. As shown in the previous Figure 2, there are two required quality control points: **(1) after generating knowledge triplets**, and **(2) after generating images**. In practice, to reduce the workload for annotators at the first point, we subdivide it into three steps: (i) after extracting knowledge components, (ii) after generating context pairs, and (iii) after generating knowledge triplets. Below, we detail the guidelines applied at each stage.

## B.1 HUMAN ANNOTATORS

Our annotation team consists of four Ph.D. students majored in computer science. All annotators are compensated for their contributions.

## B.2 FILTERING KNOWLEDGE TRIPLETS

**Filtering knowledge components.** This step removes semantically abstract phrases in the Subject, Action, and Place categories, which are difficult to visualize. Annotators classify each phrase based on its concreteness:

- **Concrete (1)**. The phrase represents tangible objects or actions.
- **Abstract (0)**. The Phrase represents abstract ideas or concepts.

Table 4 provides some labeling examples at this step:

In this step, the annotators perform labeling starting from the top 1000 most frequent phrases from each category in descending order, and keep the first 100 Subject phrases and the first 150 Action and Place phrases that meet the requirements.

Table 4: Labeling examples for knowledge components.

| Category | Label | Phrases |
|---|---|---|
| Subject | 0 | the statement, the story, the stock market, words |
| Subject | 1 | a patient, kids, politicians, whales, a sailor |
| Action | 0 | understand the event, lose weight, take care |
| Action | 1 | take a shower, use a computer, play chess |
| Place | 0 | in the event, in the newspaper, in your life |
| Place | 1 | on the ground, on the sidewalk, by the fire |

**Filtering context pairs.** The target of this step is to remove the unsatisfying phrases that remain in the candidate list after the automatic pre-filtering. To this end, our annotators conduct a binary classification on the context pairs based on their commonness in the real world. Specifically, the annotators are asked to examine the context tuples **without** the knowledge of their co-occurrence scores and label them as:

- **Common (1)**. The context tuple depicts a typical scene in the real world.

- **Uncommon (0)**. The context tuple depicts an unusual scene in the real world.

To better illustrate, we provide some labeling examples from our human annotators in Table 5 below:

Table 5: Labeling examples for context pairs.

| Category | Label | Phrases |
|---|---|---|
| (Subject, Action) | 0 | (the cat, walking the dog), (a butcher, playing cards) |
| (Subject, Action) | 1 | (the cat, climbing a tree), (a butcher, slaughtering a pig) |
| (Subject, Place) | 0 | (cows, on the beach), (a gardener, in the desert) |
| (Subject, Place) | 1 | (cows, on a farm), (a gardener, in a greenhouse) |

In this step, our annotators thoroughly label candidate context pairs within the Top K range provided by the pipeline. Context pairs labeled as **1** are retained for the next stage of generation.

**Filtering knowledge triplets.** Following the preliminary filtering steps, the workload for this phase is significantly reduced. The goal here is to eliminate any unexpected triplets that are not captured in earlier filtering. Without access to the exact co-occurrence score between the context and target phrases, annotators perform a binary classification based on the commonness of the triplet's expression. Specifically, the annotators are asked to label each triplet in the candidate list as:

- **Common (0)**. The context tuple depicts a typical scene in the real world.

- **Uncommon (1)**. The context tuple depicts an unusual scene in the real world.

Again, we provide some labeling examples from our human annotators in Table 6 for a better illustration:

Table 6: Labeling examples for knowledge triplets.

| Category | Label | Phrases |
|---|---|---|
| ((Subject, Place), Action) | 0 | ((the waitress, at the bar), using a vcr) |
| ((Subject, Place), Action) | 1 | ((the waitress, at the bar), hitting a deer) |
| ((Subject, Action), Place) | 0 | ((politicians, drinking alcohol), at the theater) |
| ((Subject, Action), Place) | 1 | ((politicians, drinking alcohol), in the oven) |

In this step, our annotators label all the candidate context-target triplets in the Top M range provided by the pipeline, and triplets with label **1** are used for subsequent image generation.

## B.3 FILTERING IMAGES

The objective of this step is to select high-quality images that align closely with the text prompt. Each annotator is provided with an image and its corresponding text query, and evaluates the image based on two criteria:

- **Alignment with text prompt (0 or 1)**. The image should contain the key objects, actions, and background described in the text. The main focus should clearly represent the scene without ambiguity or misinterpretation.
- **Image quality (0 or 1)**. The image should be clear, free from significant distortions, artifacts, or unnatural effects like warping, blurring, or pixelation.

With these two guidelines, the annotators extensively label all the generated images, and the images with a total score of **2** are kept in the final dataset.

Below, we demonstrate some sample images that are below our quality standard:

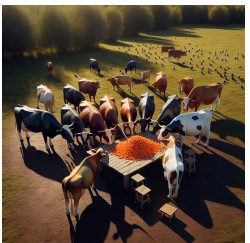 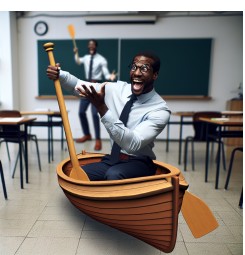 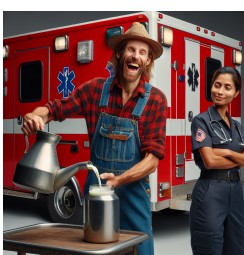 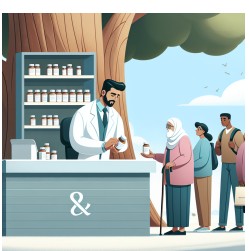

(a) Cows in a field chopping carrots.
Labels: A(0), Q(1)

(b) The teacher in a classroom paddling the boat.
Labels: A(1), Q(0)

(c) The farmer pouring milk in an ambulance.
Labels: A(0), Q(1)

(d) The pharmacist serving customers on a tree.
Labels: A(1), Q(0)

Figure 11: Examples of rejected images and their labels. **A**: Alignment with text prompt, **Q**: Quality of the image.

In this stage, we generate about 830 images in total and only keep 374 images, resulting in an average filter rate around 55%.

## C GUIDELINES FOR MANUAL LABELING OPEN-END RESPONSES

To label the correctness of MLLMs responses for the open-ended questions, we design our evaluation criteria following Xu et al. (2024a). Specifically, the annotators are asked to determine the correctness of MLLM's responses based on two criteria:

- **Relevancy (0 or 1)**: The content of the response is reflected in the image and addresses the focus of the question.
- **Responsiveness (0 or 1)**: The response directly answers the question as instructed.

Responses with a total score of **2** are considered correct for accuracy calculation.

To classify responses as either knowledge-based or vision-based, annotators grade the semantic closeness between the current response and the reference responses on a scale from 0 to 2:

- **0**: The answers are entirely unrelated.
- **1**: The answers share similar concepts but are not identical or synonymous.
- **2**: The answers are identical or synonymous.

The candidate response is assigned to the category with a higher score (e.g., Vision or Knowledge). If both scores are 0, the response is categorized as "Other".

# D ADDITIONAL EXPERIMENT RESULTS

## D.1 SUMMARY OF MLLMS PERFORMANCE ON CONFLICTVIS AND GENERAL BENCHMARKS

Table 7: Accuracy of MLLMs on general multimodal benchmarks and our CONFLICTVIS . "Text": number of text tokens, "Image": number of image-text pairs. MMB denotes MMBench developed by Liu et al. (2023c), $M^3U$ denotes MMMU by Yue et al. (2024).

| MLLMs | | Pretrain Data | | General | | Our Benchmark | | | |
|---|---|---|---|---|---|---|---|---|---|
| **Name** | **Size** | **Text** | **Image** | **MMB** | **$M^3U$** | **Y-N** | **MC** | **Subj.** | **Avg.** |
| *Open*-Source MLLMs | | | | | | | | | |
| Qwen-VL | 9.6B | 2.4T | 3.6B | 32.2 | – | 52.9 | 82.4 | 85.3 | 73.5 |
| Qwen-VL-*Chat* | | | | 61.8 | 35.9 | 69.8 | 80.5 | 89.3 | 79.9 |
| BLIP-2 | 12B | 0.2T | 0.6B | – | 35.4 | 39.3 | 82.9 | **88.8** | 70.3 |
| InstructBLIP | 13B | | | – | 35.7 | 68.2 | 88.8 | 81.8 | 79.6 |
| LLaVA-1.5 | 13B | 2.0T | 0.4B | 69.2 | 36.4 | 70.6 | 88.0 | 82.9 | 80.5 |
| LLaVA-NeXT | 8B | 15.0T | 0.4B | 71.7 | 41.7 | 70.3 | 86.6 | 85.0 | 80.7 |
| | 34B | – | 0.4B | **81.1** | **51.1** | **73.3** | **92.5** | 88.0 | **84.6** |
| *Close*-Source MLLMs | | | | | | | | | |
| Claude-3.5-Sonnet | – | – | – | 79.7 | 68.3 | 56.4 | 90.6 | 96.8 | 81.3 |
| GPT-4o | – | – | – | **83.4** | **69.1** | **74.9** | **97.1** | **97.9** | **89.9** |

# E FAILURE CASES

## E.1 CONFLICTVIS FAILURE CASES

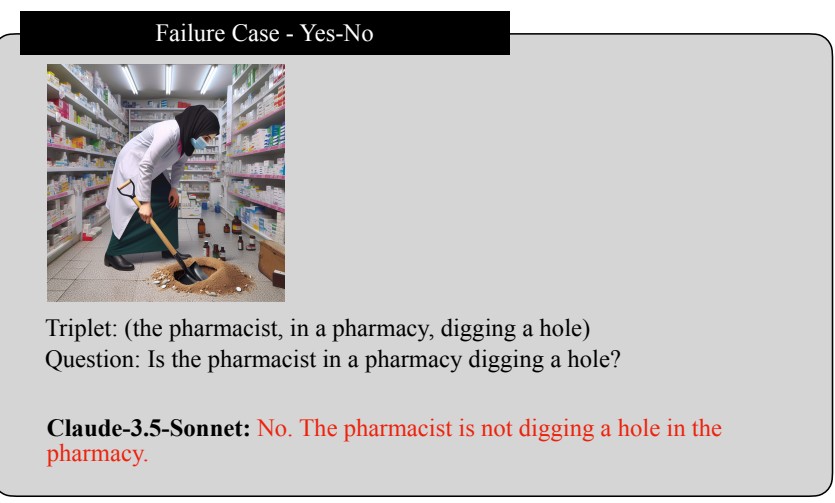

Figure 12: Failure case on yes-no question.

---

**Failure Case - Multiple Choice**

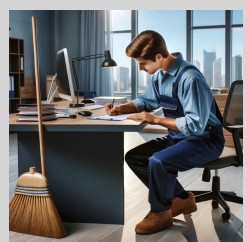

Triplet: (the janitor, in an office, signing a bill)
Question: What is a janitor doing in an office?
(A) sweeping the floor   (B) eating the roadrunner
(C) honoring a knight    (D) signing a bill

**GPT-4o:** (A) sweeping the floor

Figure 13: Failure case on multiple-choice question.

---

**Failure Case - Subjective**

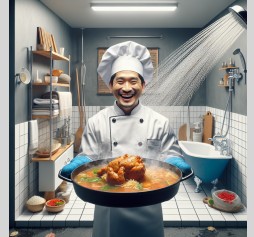

Triplet: (a cook, serving food, in a shower)
Question: Where is a cook serving food?

**LLaVA-NeXT-34B:** Kitchen

Figure 14: Failure case on subjective question.

E.2  CoT FAILURE CASES

**CoT Failure Case - Knowledge Over-reliance**

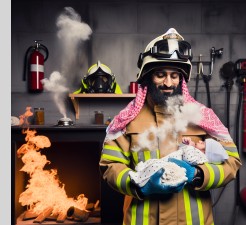

Triplet: (the fireman, powdering the baby, by the fire)
Question: What is the fireman doing by the fire?

*Original (zero-shot):*

**Qwen-VL-Chat:** The fireman is standing in front of the fire and holding a baby.

*CoT:*

**Qwen-VL-Chat:** *Let's think step by step.* We can see a fireman in the picture. We know that firemen put out fires, so the fire in the picture must be put out by the fireman. Therefore, the answer is "put out the fire".

Figure 15: CoT failure mode 1: The MLLM responses according to its commonsense knowledge.

**CoT Failure Case - Denial**

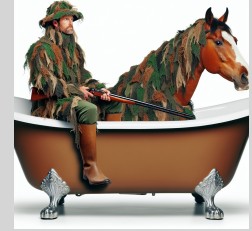

Triplet: (the hunter, riding a horse, in a bathtub)
Question: Where is the hunter riding a horse?

*Original (zero-shot):*

**LLaVA-NeXT-34B:** Bathtub.

*CoT:*

**LLaVA-NeXT-34B:** *Let's think step by step.* The image shows a man dressed in camouflage clothing, which is often associated with hunters. He is holding a rifle, which is a common hunting tool. However, instead of riding a horse, he is sitting in a bathtub. This is an unusual and humorous juxtaposition, as one would typically expect to see a horse in a natural setting rather than a bathtub. Therefore, the answer is: Not possible in this image.

Figure 16: CoT failure mode 2: The MLLM asserts that answering the question is impossible.

