# OpenReview forum: "Insight Over Sight? Exploring the Vision-Knowledge Conflicts in Multimodal LLMs"
_ICLR.cc/2025/Conference — ICLR 2025 Conference Withdrawn Submission_

### Official Review · Reviewer_pQpV · 2024-10-28

**Soundness:** 2
**Presentation:** 3
**Contribution:** 2
**Rating:** 3
**Confidence:** 4

**Summary:**

The paper addresses vision-knowledge conflicts in multimodal large language models (MLLMs), where the model's commonsense knowledge can conflict with visual inputs, often leading to vision-irrelevant outputs. To tackle this, the authors introduce an automated pipeline and a new benchmark, ConflictVis, designed to create scenarios that test MLLMs on handling commonsense contradictions between text and visual information. The study shows that MLLMs frequently over-rely on parametric knowledge, especially for simpler questions, and introduces a "Focus-on-Vision" (FoV) prompting strategy to encourage MLLMs to prioritize visual data. Experimental results across nine models indicate that the FoV strategy enhances performance by reducing dependency on textual information in visually conflicting contexts.

**Strengths:**

- Offers a novel and well-defined benchmark (ConflictVis) with rigorous human-in-the-loop validation.
- Good experiment setups including sanity check, comprehensive question type evaluation.
- It's a well-structured, well-written, and easy-to-follow paper.

**Weaknesses:**

- The practical relevance of these visual conflict scenarios in real-world applications is unclear. I don't think users would actually input counter-commonsense images, such as a baby on a bed fixing a computer in their daily lives. I would recommend using use-cases in WildVision[1] which collects real-world use-cases. Additionally, the reliance on a benchmark that emphasizes rare, contrived scenarios may not reflect typical user interactions with MLLMs, potentially limiting the benchmark’s broader applicability in evaluating MLLM performance.
- In comparison to textual knowledge conflicts, the memorization effect here is relatively low and can be addressed with a simple prompt strategy, which reduces the significance of this issue.
- The proposed FoV method, though effective, is a simple prompt adjustment that may not generalize across all multimodal contexts or complex use cases beyond commonsense conflicts. In fact, for all multimodal inputs, it seems intuitive that prompts should at least include “Based on the given image.” The limited utilization of visual information could be a result of poorly structured initial prompts used in Section 3.3. (Incidentally, what was the exact prompt in Section 3.3?). Thus, the degree of knowledge conflict may not be as serious as suggested by the authors.

Overall, I have concerns on the generalizability of the findings and the practicality of the benchmark scenarios. While the work provides useful insights into handling vision-knowledge conflicts, the proposed solutions and evaluation settings may not align well with real-world usage or fully address the complexities of multimodal reasoning in MLLMs.

[1] Lu et al., WildVision: Evaluating Vision-Language Models in the Wild with Human Preferences, NeurIPS 2024.

**Questions:**

what was the exact prompt in Section 3.3?

---

### Official Review · Reviewer_M694 · 2024-11-01

**Soundness:** 3
**Presentation:** 2
**Contribution:** 2
**Rating:** 5
**Confidence:** 4

**Summary:**

This paper studies the context-memory knowledge conflicts in MLLMs by constructing a counter-commonsense multimodal benchmark.
They generate images using less frequent commonsense triplets.
The results show that MLLMs have problems when facing counter commonsense visual information.
They also design a prompting strategy to mitigate the problem.

**Strengths:**

1. The paper studies an overlooked problem of vision-knowledge conflicts for MLLMs.

2. The paper's generated images serve as a contribution to constructing counter-commonsense conflicts.

**Weaknesses:**

1. The proposed benchmark does not fully capture the severity of vision-knowledge conflicts, as GPT-4 achieves more than 90% accuracy, suggesting that more challenging scenarios might be necessary to evaluate SOTA models.

2. The analysis of vision-knowledge conflicts remains relatively superficial.
The fundamental reason stated in the paper can be attributed to a long-standing common opinion that MLLMs have language bias, which is already pointed out by previous works.

3.  This work only investigate the counter-commonsense conflicts and does not explore other types of vision-knowledge conflicts, such as those involving factual conflicts and world-knowledge conflicts.

**Questions:**

1. Can you clarify why model accuracy remains high on this dataset?
Although you describe this task as more challenging than traditional VQA, the performance does not show a significant gap between them.
If the task could be more challenging, there might be more to analyze.
For now, the cause of this phenomenon can be easily attributed to the language bias because MLLMs rely more on the textual modality.
If you can introduce more diverse conflicts, you might be able to find out new problems in MLLMs.


2. Why do you use the vicuna-13b for probability rather than larger or more powerful model.
To ensure the commonsense is embedded in the model, would it be better to train a model using a commonsense corpus?

3. Could you explain why Yes/No questions have lower performance compared to other question types?

---

### Official Review · Reviewer_WBPW · 2024-11-04

**Soundness:** 2
**Presentation:** 2
**Contribution:** 2
**Rating:** 3
**Confidence:** 4

**Summary:**

This paper studies knowledge conflicts in multimodal large language models (MLLMs). The authors propose a human-in-the-loop pipeline to create ConflictVis, a benchmark designed to elicit knowledge conflicts, comprising 374 examples. Each example consists of a generated image and four questions. The authors use ConflictVis to test nine MLLMs and show models overly-rely on textual inputs as well as their parametric knowledge. Finally, they propose Focus-on-Vision (i.e., the prompt “Please focus on the visual information”) to counter underutilization of visual information.

**Strengths:**

1. The topic is timely and important
2. Once the issues below are addressed, ConflictVis can be a useful benchmark to test models.

**Weaknesses:**

In general, I find the premise of the paper to be good and interesting: Detecting knowledge conflicts or visual information underutilization is important and the structure of instances in ConflictVis is easy to understand. However, It is very much not clear to me why some questions are harder than others nor why these are the right questions to ask about the images. In section 3, the text omits a lot of detail and as a result, the conclusions are not convincing.

These weaknesses can be improved, but require major overhaul to section 3, and possibly to section 2.

1. The paper jumps between textual context (i.e., the input text) and parametric knowledge, i.e., the information the model encodes, irrespective of any particular input. Sometimes it refers to them both as “underutilization of visual information”, which perhaps would have been the approach to take throughout the entire paper. But it doesn’t take the time to clearly distinguish between them, which makes it hard to follow (lines 473-499 shortly makes this distinction, but it is missing from the rest of the paper).


2. **ConflictVis**
- From my understanding, the method requires human evaluation every time it is used (lines 277-278). It is not clear to me why. Especially if the images are not the subject of evaluation, then the correct answers can be predetermined and be marked as part of ConflictVis. This way, you would only need an LLM to compare the output by the MLLM with the predetermined answer, and this step would be automatic.
- There is no detail about what makes some questions harder than others, or why multiple difficulties are needed.


**Substantial Details Missing in Experiments**
- **Section 3.2 Clarity on MLLMs Output Comparison:**
  - The paper does not specify what the MLLMs outputs are compared against in the sanity test. It is assumed to be the yes/no answers from human annotations, but this is not explicitly stated.
    - **Suggestion:** Clarify the comparison benchmarks in the text or provide a reference to where those details can be found.

- **Uncertainty Calculation and Aggregation (lines 301-314):**
  - The method of computing and aggregating uncertainty is not described. Additionally, details such as the number of samples from each benchmark, how these samples were selected, and the sampling method are absent.
    - **Suggestion:** Include a more detailed methodology or direct the reader to an appendix where these methods are outlined.

**Clarity on MLLM Responses in Section 3.3 Without Images:**
- **Context of Questions Without Images:**
  - It is unclear what kind of answers the MLLMs provide to questions containing determiners when no image is present to define the referent. For instance, the question posed in line 295, "Is *the baby* on the bed fixing a computer?", assumes knowledge of 'the baby' which hasn't been introduced.
    - **Potential Issue:** If an MLLM like GPT-4o rejects this question due to the lack of contextual introduction of 'the baby', and this is counted negatively, it suggests a design flaw in the experiment. The experiment should distinguish between a model's reliance on introduced contextual knowledge versus its parametric knowledge.
    - **Suggestion:** Clarify how responses are evaluated in the absence of images and consider revising the methodology to accurately test for over-reliance on textual versus visual information. This could involve a different scoring approach where the context provided by images is factored into the evaluation of responses.

**Critique of Focus on Vision (FoV) Methodology**
- **Inconsistency and Lack of Improvement (Table 2):**
  - The FoV approach, which merely prompts the model to focus on visual information, does not introduce a novel technique, as implied in the abstract and introduction. The data presented in Table 2 does not demonstrate consistent or meaningful improvement over the existing baselines. When improvements do occur, they are marginal.
    - **Implication:** If FoV had shown a significant performance gap over other baselines, it could have substantiated the paper's claims about the under-utilization of visual information in current models.

**Questions:**

1. Why are the open-ended questions called “subjective”? They do not appear to be subjective at all. For example, Figure 9 shows a person with a paddle in his hands. Why is “playing a guitar” a subjective answer if it is objectively not true? Similarly, in figure 14, why is the answer to “where is a cook serving food?” subjective? It is quite clear the cook is standing in a bathroom.

2. Section 3.4, what are the hyperparameters to reproduce the results in table 2?


**Other**

* The human-in-the-loop components in Figure 2 are not clear.
As a note, it would have been nice if there was an attempt to understand why MLLMs underutilize visual information, if you believe this is the case, but I believe the resource itself can be useful as is.

---

### Note · Authors · 2024-11-15

I have read and agree with the venue's withdrawal policy on behalf of myself and my co-authors.